# Anti-Colitic Effects of Ethanol Extract of *Persea americana* Mill. through Suppression of Pro-Inflammatory Mediators via NF-κB/STAT3 Inactivation in Dextran Sulfate Sodium-Induced Colitis Mice

**DOI:** 10.3390/ijms20010177

**Published:** 2019-01-05

**Authors:** Joo Young Hong, Kyung-Sook Chung, Ji-Sun Shin, Geonha Park, Young Pyo Jang, Kyung-Tae Lee

**Affiliations:** 1Department of Pharmaceutical Biochemistry, College of Pharmacy, Kyung Hee University, 26, Kyungheedae-ro, Dongdaemun-gu, Seoul 02447, Korea; hjisuk1206@naver.com (J.Y.H.); adella76@hanmail.net (K.-S.C.); jsshin@khu.ac.kr (J.-S.S.); 2Department of Life and Nanopharmaceutical Sciences, Graduate School, Kyung Hee University, 26, Kyungheedae-ro, Dongdaemun-gu, Seoul 02447, Korea; ginapark0326@khu.ac.kr (G.P.); ypjang@khu.ac.kr (Y.P.J.); 3Department of Oriental Pharmaceutical Science, College of Pharmacy, Kyung Hee University, 26, Kyungheedae-ro, Dongdaemun-gu, Seoul 02447, Korea

**Keywords:** *Persea americana* Mill., avocado, DSS, macrophage, anti-inflammation, NF-κB, STAT3

## Abstract

*Persea americana* Mill, cv. Hass, also known as avocado, has been reported to possess hypolipidemic, anti-diabetic, anti-oxidant, cardioprotective, and photoprotective potencies. However, few studies have reported its anti-colitic effects. In this study, we investigated anti-colitic effects of ethanol extract of *P. americana* (EEP) in dextran sulfate sodium (DSS)-induced colitic mice and the involved molecular mechanisms. EEP effectively improved clinical signs and histological characteristics of DSS-induced colitis mice. In DSS-exposed colonic tissues, EEP reduced expression levels of inducible nitric oxide synthase (iNOS), cyclooxygenase-2 (COX-2), and pro-inflammatory cytokines such as interleukin (IL)-6, IL-1β, and tumor necrosis factor (TNF)-α. Moreover, EEP suppressed DSS-induced activation of nuclear factor-kappa B (NF-κB) and signal transducer and activator of transcription 3 (STAT3). Consistent with in vivo results, EEP also suppressed protein and mRNA expression levels of iNOS, COX-2, and pro-inflammatory cytokines via NF-κB and STAT3 inactivation in LPS-induced RAW 264.7 macrophages. Taken together, our data indicate that ethanol extract of avocado may be used as a promising therapeutic against inflammatory bowel diseases by suppressing the NF-κB and STAT3 signaling pathway.

## 1. Introduction

Inflammatory bowel diseases (IBD), covering Crohn’s disease (CD) and ulcerative colitis (UC), are characterized by chronic remittent or progressive inflammatory conditions that can damage the entire gastrointestinal tract and the colonic mucosa [1]. The most common age for IBD is 20 to 29 years and the second most common age is 50 to 70 years [2]. Although the exact etiology of IBD is not clearly understood yet, accumulating reports indicate that abnormal immune responses, complex gene–environment interactions, and genetic factors can converge to provoke disease initiation and progression [3]. Clinically, 5-aminosalicylic acid (5-ASA), corticosteroids, sulfasalazine, and biological agents such as adalimumab are used to treat UC. However, the side effects of many drugs are still remained [4]. Recently, a variety of natural products that can safely regulate pro-inflammatory signals and modulate IBD have been reported [5].

Inflammation is caused by the host’s immune response to pathogens or tissue injuries, and mediated by cytokines [6]. Primary inflammatory stimuli such as microbial products and cytokines act through specific receptors and cause changes in gene expression by activating various intracellular signaling pathways, leading to activation of transcription factors such as nuclear factor-кB (NF-κB) and signal transducers and activators of transcription 3 (STAT3) [7]. Since these transcription factors are markedly induced in many IBD patients, they have become targets of various therapeutic agents [8,9].

*Persea americana*, also called avocado, is a berry fruit, which has dark green leathery skin and a large seed. As the interest of people for avocado increases, the number of papers published is increasing (Appendix A). Among many varieties (e.g., Fuerte, Hass, Pinkerton, Bacon, Azteca, Ettinger, or Rincon), Hass avocado is the most abundant avocado cultivar in the world [10]. This fruit contains a variety of essential nutrients and important phytochemicals such as, lutein, zeaxanthin, vitamin E, and phenolic antioxidants [11]. There are many reports about health effects of avocado, including its beneficial effects on total cholesterol and body weight control, promoting heart healthy lipid profiles, and managing hypercholesterolemia [12,13,14]. Furthermore, polyhydroxylated fatty alcohols derived from avocado may help to protect the skin from ultraviolet-induced inflammatory damage [15]. It has been reported that arthrocen consisting of avocado and soy unsaponifiable extracts in a 1:2 ratio has protecting effects on osteoarthritis through down-regulating inflammatory responses in chondrocytes [16]. However, protective effects of *P. americana* on intestine inflammation and related mechanisms are currently unknown. Therefore, the objective of the present study was to examine whether ethanol extract of *P. americana* (EEP) could suppress inflammatory responses in dextran sodium sulfate (DSS)-induced colitis mice model and LPS-induced macrophages. Mechanisms involved in such effects were also explored through biochemical evaluations.

## 2. Results

### 2.1. Identification of Phytochemicals in EEP

The HPLC chromatogram of the ethanol extract of avocado (*P. americana* Mill., Lauraceae) was monitored at 280 nm. Results are shown in Figure 1. Five compounds were identified by HPLC-PDA-ESI-MS by comparing their UV spectra and high-resolution mass number with literature data previously reported. Peak 1 was confirmed as homocysteine [17] while peak 2, peak 3, and peak 5 were identified as guanine, cytidine, and guanosine, respectively [18]. Peak 4 was verified as uridine [19]. Retention time, molecular weight of precursor ion, and molecular formula of each compound are listed in Table 1. Results showed that major characteristic metabolites of 30% ethanol extract of avocado seemed to be nucleobase and nucleosides. The optimized HPLC chromatogram and the profiling of major metabolites of this extract could be utilized to assure the consistency and reproducibility of biological responses of the extract. In addition, EEP has total polyphenol and total flavonoid contents of 4.52 ± 0.01 mg/g and 1.56 ± 0.05 mg/g, respectively (Table 2).

### 2.2. Effects of EEP on Symptoms of DSS-Induced Colitis Mice

To investigate the anti-colitis potential of EEP in vivo, we used DSS-induced colitis mice model. Mice were administrated 4% DSS in drinking water for 9 days with or without EEP orally (50, 100, or 200 mg/kg/day, p.o.). As a positive control, 5-ASA (75 mg/kg/day) was used. Following induction of colitis, combinatorial disease activity index (DAI), integrated from body weight loss, stool consistency, and gross bleeding were increased in DSS only-treated mice. However, administration of EEP dose-dependently reduced DAI values compared to DSS only-treated group (Figure 2A). DSS administration induced colon inflammation, and macroscopic colon damage with reduction in colon length. Colon length of the DSS only-treated group was significantly shorter than that of vehicle-treated control group (9.90 ± 0.77 cm vs. 6.01 ± 0.65 cm, *p* < 0.001). In DSS plus EEP (200 mg/kg)-treated group, shortening of the colon length was significantly recovered (6.01 ± 0.65 cm vs. 7.47 ± 0.25 cm, *p* < 0.01). DSS plus EEP (50 or 100 mg/kg)-treated group showed mild recovery for the shortening of colon length induced by DSS (Figure 2B). A variety of symptoms including splenomegaly may occur at very early onset of IBD to warn the risk of disease [20,21]. In the present study, spleen index (spleen weight (g)/body weight (kg)) of the DSS only-treated group was higher than that of the vehicle-treated control group (9.76 ± 0.92 g/kg vs. 3.91 ± 0.59 g/kg, *p* < 0.001). Treatment with DSS plus EEP significantly prevented spleen hypertrophy in mice with DSS-induced colitis in a dose-dependent manner (EEP 50 mg/kg: 7.99 ± 0.51 g/kg, *p* < 0.05, EEP 100 mg/kg: 6.54 ± 0.74 g/kg, *p* < 0.05, EEP 200 mg/kg: 5.53 ± 0.84 g/kg, *p* < 0.01) (Figure 2C). These data suggest that administration of EEP could undermine symptoms of DSS-induced colitis.

### 2.3. Effects of EEP on Histological Changes in DSS-Induced Colitis Mice

To confirm these beneficial effects, pathological evaluation was performed by light microscopic examination of colonic segments after hematoxylin and eosin (H&E) staining. The colon tissue from DSS only-treated group showed evidence of mucosal damage, ulceration, and crypt distortion. However, colon sections from DSS plus EEP (50, 100, or 200 mg/kg)-treated groups showed decreases in pathological signs of tissue damage with less cryptic injury (Figure 3A). In DSS only-treated group, crypt length was significantly decreased by 67.58 ± 7.71% compared to that of vehicle-treated control group (100 ± 7.01%). However, DSS plus EEP at 100 mg/kg and 200 mg/kg significantly restored the shortened crypt length by 78.21 ± 9.31% (*p* < 0.01) and 99.97 ± 7.12% (*p* < 0.001), respectively (Figure 3B). Next, we analyzed tissue apoptosis and inflammatory cell infiltration as pathological features of IBD [22]. Whole proteins obtained from tested colonic tissues were analyzed for expression levels of apoptotic marker proteins by Western blotting. Increased levels of cleaved poly (ADP ribose) polymerase (PARP) in colonic tissues demonstrated that apoptotic colonic cells were abundant in DSS only-treated group. However, treatment with EEP (50, 100, or 200 mg/kg) significantly decreased expression levels of cleaved PARP, demonstrating that EEP could protect the intestine epithelium against DSS-induced apoptosis (Figure 3C). We also evaluated neutrophil infiltration into colon tissue by determining activities of myeloperoxidase (MPO), a pro-inflammatory enzyme stored in neutrophilic granulocytes. As shown in Figure 3D, DSS exposure significantly enhanced MPO activity in colon tissue whereas EEP treatment (50, 100, or 200 mg/kg) significantly decreased DSS-induced MPO activity. To further confirm macrophage infiltration, we measured mRNA expression levels of F4/80, a major marker of macrophage [23]. Compared to the vehicle-treated control group, the DSS only-treated group showed significant increase in F4/80 mRNA expression. However, treatment with DDS plus EEP (50, 100, or 200 mg/kg) restored altered expression levels of F4/80 (Figure 3E).

### 2.4. Effects of EEP on mRNA Expression of Pro-Inflammatory Cytokines in DSS-Induced Colitis Mice

Pro-inflammatory cytokines, small secreted proteins released by cells, are involved in the up-regulation of inflammatory reactions and immune systems [24]. They also contribute to the perpetuation of colonic inflammation [25]. To investigate mucosal inflammatory states, mRNA expression levels of interleukin (IL)-6, IL-1β, and tumor necrosis factor (TNF)-α in colonic tissues were measured by RT-PCR. Treatment with DSS for 9 days markedly increased mRNA expression levels of these cytokines. However, DSS plus EEP treatment at 50, 100, or 200 mg/kg significantly suppressed expression levels of pro-inflammatory cytokines induced by DSS (Figure 4).

### 2.5. Effects of EEP on Protein Expression of Inducible Nitric Oxide Synthase (iNOS) and Cyclooxygenase (COX)-2 and the Activation of NF-κB and STAT3 in DSS-Induced Colitis Mice

Next, we investigated molecular mechanisms involved the effect of EEP on DSS-induced colitic tissue. Development of ulcerative colitis is known to be accompanied by the activation of COX-2 and iNOS followed by increased production of prostaglandin (PG) E_2_ and nitric oxide (NO) in colonic tissues [26]. Although expression levels of COX-2 and iNOS were clearly upregulated in DSS only-treated group, DSS plus EEP-treated group showed significantly suppressed expression of COX-2 and iNOS (Figure 5A). Expression of COX-2, iNOS, and pro-inflammatory cytokines is known to be regulated by STAT3 and NF-κB [27,28]. Thus, we examined the phosphorylation of p65 NF-κB subunit and STAT3 in colonic tissues via Western blotting. Treatment with DSS significantly increased phosphorylation levels of p65 and STAT3, whereas these increases were markedly attenuated by treatment with DSS plus EEP at 50, 100, or 200 mg/kg (Figure 5B). These effects were further confirmed by immunohistochemical analysis. As shown in Figure 5C, DSS plus EEP treatment at 200 mg/kg suppressed DSS-induced phosphorylation of p65 and STAT3.

### 2.6. Effects of EEP on LPS-Induced Production of NO and PGE_2_ by Suppressing Expression of iNOS and COX-2 in RAW 264.7 Macrophages

Macrophages are major leukocytes in intestinal mucosa and involved in IBD through secreting pro-inflammatory cytokines under inflammatory conditions [29]. Therefore, we additionally performed experiments to determine anti-inflammatory effects of EEP using RAW 264.7 murine macrophages. Effects of EEP on the production of inflammatory mediators including NO and PGE_2_ were determined. Our results revealed that treatment with EEP at 100 or 200 μg/mL significantly suppressed LPS-induced NO production. Such inhibitory effects of EEP were not caused by its nonspecific cytotoxicity because EEP at concentration up to 200 μg/mL showed no effect on cell viability based on 3-(4,5-Dimethylthiazol-2-yl)-2,5-diphenyl-tetrazolium bromide (MTT) assay (Figure 6A). PGE_2_ production was also significantly inhibited by treatment with EEP at 200 μg/mL (Figure 6B). Protein and mRNA expression levels of iNOS and COX-2 were then determined by Western blotting and qRT-PCR analysis, respectively. Treatment with EEP dose-dependently downregulated protein and mRNA expression levels of iNOS and COX-2 in LPS-induced RAW 264.7 macrophages (Figure 6C,D).

### 2.7. Effects of EEP on LPS-Induced Production and mRNA Expression Levels of Pro-Inflammatory Cytokines in RAW 264.7 Macrophages

To examine the suppressive effect of EEP on pro-inflammatory cytokines, we determined both production and mRNA expression levels of IL-6, IL-1β, and TNF-α in LPS-induced macrophages. Treatment with EEP significantly decreased LPS-induced production of IL-6, IL-1β, and TNF-α in a concentration-dependent manner (Figure 7A–C). IL-6 and IL-1β mRNA expression levels were also significantly decreased by treatment with EEP at 50, 100, or 200 μg/mL while mRNA expression level of TNF-α was significantly decreased by EEP at 200 μg/mL (Figure 7D,F). These results indicate that EEP can suppress expression of these inflammatory cytokines at the transcriptional level.

### 2.8. Effects of EEP on Activation of NF-κB and STAT3 in RAW 264.7 Macrophages

Activation of NF-κB and STAT3 is critically required for LPS-induced transcriptional regulation of inflammation [30,31]. Thus, we examined the effect of EEP on LPS-induced phosphorylation of p65 (subunit of NF-κB) and STAT3 by Western blotting. Our results showed that EEP attenuated LPS-induced phosphorylation of p65 and STAT3 in a concentration-dependent manner (Figure 8). Collectively, these results suggest that anti-colitis effects of EEP are caused by NF-κB and STAT3 inactivation both in vivo and in vitro.

## 3. Discussion

Although inflammation is a protective response of host immunity, aberrant inflammatory responses can lead to host tissue injuries [32]. IBD are a group of inflammatory disorders affecting the gastrointestinal tract. They are accompanied by a range of symptoms due to inflammation of the gut, including abdominal pain, fever, vomiting, diarrhea, rectal bleeding, anemia, and weight loss [33]. Since patients with IBD are at significantly increased risk of colorectal cancer, epidemiologic studies have shown increased risk of colorectal cancer in IBD [34]. Since established pharmacological agents have side effects, patients with IBD want to know if there are any non-pharmacological treatments or precautions and what they should eat [4]. For this reason, the importance of diet in treatment and prevention of IBD is increasing. Many studies have been done, leading to good classification for what to eat and what not to eat in state of IBD [35]. According to one dietary strategy, low-fermentable oligosaccharide, disaccharide, monosaccharide, and polyol diets (low-FODMAPs) are recommended to help patients manage their functional gut symptoms while avocado should be avoided [36]. However, our findings revealed the preventive effect of EEP against DSS-induced colitis. DSS-induced colitis model has been widely used due to its simplicity and many similarities with human ulcerative colitis [37]. In the present study, DSS administration induced typical features of colitis such as body weight loss, diarrhea, bloody stool, and colon shortening in DSS-induced colitis mouse model. However, treatment with EEP significantly recovered these clinical signs.

The five compounds of EEP, identified by HPLC-PDA-ESI-MS, are nucleosides and nucleotides which participate in many cellular functions, including inflammation regulation by acting as signaling molecules [38]. Homocysteine has been reported to reveal anti-inflammatory properties in a hypercholesterolemic rat model [39]. And uridine has been demonstrated to reduce inflammatory cell infiltration and cytokine production in sephadex-induced lung inflammation [38]. Guanosine exhibits anti-inflammatory activity by associating the heme oxygenase-1 (HO-1) signaling hippocampal astrocytes [40]. Additionally, we examined the total polyphenol and total flavonoid contents of EEP to support the anti-inflammatory activity of EEP, because polyphenol and flavonoid were well known to their anti-inflammatory activity [41]. These reports suggested that the complex of these compounds from EEP is contributed to anti-colitic and anti-inflammatory properties of EEP.

Pathological features of IBD are closely linked to exaggerated inflammation and consequential destruction of intestinal epithelium [42]. Epithelium damage during IBD occurs through apoptosis mechanisms. This has been clinically observed in the intestinal epithelium of patients [43]. Thus, we detected cleaved form of PARP, one of DNA repair enzymes, as a marker of apoptosis in DSS-induced colitis model [44]. Under IBD conditions, inflammatory cells such as neutrophils and macrophages are increased in the colon and have critical roles in disease progression by secreting many inflammatory molecules [45,46]. Although neutrophils are necessary to eliminate foreign particles and bacteria, persistent infiltration causes tissue damage through many different substances produced from neutrophils such as MPO [47]. Just like in neutrophils, macrophages remove pathogens through MPO activity [48]. It has been well-known that monocytes are recruited to the lamina propria of colon and become inflammatory macrophages during intestinal inflammation [29,49]. These macrophages damage colonic tissues by inflammatory responses with expression of iNOS, COX-2, and pro-inflammatory cytokines [26,50]. The F4/80 glycoprotein has been reported as one of the most specific cell-surface markers for murine macrophages and increases as macrophages mature [23,51]. High level of F4/80 expression can lead to enhanced ability of macrophages in adipose tissue to produce pro-inflammatory cytokines [52]. In the present study, colon tissue in DSS-treated group showed signs of damage such as crypt length reduction, PARP cleavage, and inflammatory cell infiltration as evidenced by MPO activity and mRNA expression of F4/80. However, treatment with EEP effectively ameliorated these pathological changes.

Transcription factors NF-κB and STAT3 are major regulatory components regulating the production of these cytokines and signaling molecules involved in IBD. Several studies have reported that the activation of NF-κB and STAT3 is increased in IBD patients [8,9]. NF-κB, a family of transcription factors, consists of five different subunits: p65 (RelA), c-Rel, RelB, p50, and p52 [53]. Among these NF-κB members, only p65, c-Rel, and RelB have a C-terminal transcriptional activation domain that can activate transcription directly by phosphorylation [54]. Increased NF-κB expression in mucosal macrophages is associated with upregulated capacity of these cells to produce iNOS, COX-2, and cytokines (IL-6, IL-1β, and TNF-α), thus mediating inflammatory responses [55]. STAT3 is phosphorylated and activated by several cytokines including IL-6. It then becomes dimer and translocates into the nucleus to induce expression of genes encoding molecules related to various biological functions such as cytokine production, anti-apoptosis, pro-apoptosis, and cell growth [56,57]. These transcription factor signaling pathways cooperatively mediate proliferative and anti-apoptotic activities required for oncogenesis under inflammatory conditions. Subunit of NF-κB such as p65 can interact with STAT3 and this combination can regulate their transcriptional activity. Many cytokines such as IL-6 induced by NF-κB and STAT3 can feed back to enhance their activation [58,59,60]. Effects of avocado-soybean unsaponifiables (ASU) on symptomatic osteoarthritis have been reported, showing that ASU can decrease PGE_2_ production, IκBα degradation, NF-κB nuclear translocation, and ERK1/2 activation in chondrocytes [16]. In a previous pilot study, healthy participants with hamburger patty added with 1/2 of an avocado showed reduced serum levels of IL-6 and NF-κB activation in peripheral blood mononuclear cells compared to the group with normal beef hamburger patty [61]. However, effects of avocado on STAT3 activity have not been studied sufficiently. Our findings revealed that EEP suppressed protein expression levels of inducible pro-inflammatory enzymes (COX-2 and iNOS) and mRNA expression levels of cytokines (IL-6, IL-1β, and TNF-α) due to inhibitory effects of EEP on phosphorylation of NF-κB and STAT3 in DSS-induced colitis mice. Consistent with these results, EEP inhibited the production of NO, PGE_2_, and inflammatory cytokines by suppressing phosphorylation of NF-κB and STAT3 in LPS-induced RAW 264.7 macrophage cells. Our findings indicate that EEP could be a useful pharmacologic tool to improve our understanding of cellular functions. In addition, EEP can be considered as a potential treatment option for colitic diseases.

## 4. Material and Methods

### 4.1. Chemicals and Reagents

Dulbecco’s modified Eagle’s minimum essential medium (DMEM), fetal bovine serum (FBS), penicillin, and streptomycin were purchased from Life Technologies Inc. (Grand Island, NY, USA). The antibodies for iNOS, COX-2, p65, PARP, β-actin, normal rabbit IgG and normal mouse IgG_2b_ were obtained from Santa Cruz Biotechnology Inc. (Santa cruz, CA, USA). The enzyme immunoassay (EIA) kits for PGE_2_, IL-6, IL-1β, and TNF-α were purchased from R&D Systems (Minneapolis, MN, USA). TOPscript™ RT Dry MIX was purchased from Enzynomics Co. Ltd. (Daejeon, Korea). iNOS, COX-2, TNF-α, IL-6, IL-1β, and β-actin oligonucleotide primers were purchased from Bioneer (Seoul, Korea). MTT, l-N^6^-(1-iminoethyl)lysine (l-NIL), NS-398, *Escherichia coli* LPS and all other chemicals were obtained from Sigma Chemical Co. (St. Louis, MO, USA). HPLC-grade acetonitrile was purchased from Fisher Scientific Korea (Seoul, Korea) and HPLC-grade formic acid was purchased from Wako (Osaka, Japan).

### 4.2. Preparation of Ethanol Extract of Persea americana Mill.

The powdered pulp part of Avocado (57.6429 g) was extracted three times with 30% ethanol (as 10 times concentration) with ultra-sonication. The extract was filtered with a filter paper (Hyundai Micro Co, Ltd., Seongnam, Korea) and freeze-dried to get 30% ethanol extract (11.2726 g; yield 19.6%).

### 4.3. Sample Preparation and HPLC-PDA-ESI-MS Analysis

The extract of *P. americana* Mill. was dissolved in dimethyl sulfoxide and diluted with deionized water to prepare final concentration of 50 mg/mL stock solution. The sample was filtered through a 0.2 µm polyvinylidenefluoride (PVDF) syringe filter (Whatman International Ltd., Maidstone, Kent, UK) before being injected.

A Waters Alliance 2690 HPLC system (Waters Corp., Milford, MA, USA) linked with a photodiode array detector (PDA 2996 detector, Waters Corp.) and JMS-T100TD (AccuTOF-TLC) (JEOL Ltd., Tokyo, Japan) mass spectrometer equipped with electrospray ionization (ESI) source were used for chromatographic and mass spectrometric (MS) analysis. The chromatographic separation was carried out on the Waters Atlantis^®^ T3 C18 column (250 × 4.6 mm id, 5 μm) (Waters Corp., Milford, MA, USA). The mobile phase consisted of acetonitrile (solvent A) and acidified water with 0.1% formic acid (solvent B). The gradient condition of the mobile phase was 0–5 min, 0%; 5–10 min, 0% to 10%; 10–15 min, 1%; 15–70 min, 1% to 60%; 70–71 min, 60% to 100%; 71–80 min, 100% as percent of solvent A. The injection volume was 10.0 µL. The flow rate was set at 1.0 mL/min and the column oven temperature was maintained at 25 °C and the detection wavelength was 280 nm. The conditions of MS analysis in the positive ion mode were as follows: scan range, *m*/*z* 50–1000; desolvating chamber temperature, 250 °C; orifice1 temperature, 80 °C; orifice 1 voltage, 80 V; orifice 2 voltage, 10 V; ring lens voltage, 5 V; peak voltage, 1000 V; detector voltage, 1900 V; nitrogen gas flow rate, 1.0 L/min (nebulizing gas) and 3.0 L/min (desolvating gas).

### 4.4. Measurement of Total Phenolic Content

The assay was determined using 1 mL of extract stock solution and 1 mL of each standard tannic acid (SAMCHUN Chemicals., Seoul, Korea) was taken in 50 mL volumetric flask, added 7.5 mL of water, 0.5 mL of Folin-Denis reagents (Junsei Chemical Co., Tokyo, Japan) and 1 mL of 35% sodium carbonate solution was mixed in each volumetric flask. Absorbance were measured at 760 nm by micromultiplate reader after 30 min in dark. The total phenolic content was calculated from the calibration curve of tannic acid, and the results were expressed in mg tannic acid equivalent per gram of dried EEP.

### 4.5. The Measurement of Total Flavonoid Content

The total flavonoid content was expressed as mg of total flavonoids per gram of the dried EEP after reacting for 15 min at room temperature by adding 100 μL of 2% aluminium chloride hexahydrate to 100 μL of each extract. The absorbance was detected by micromultiplate reader at 430 nm. The total flavonoid content was calculated from the standard curve of quercetin (Sigma Chemical Co., St. Louis, MO, USA).

### 4.6. Cell Culture

The RAW 264.7 murine macrophage cells (Korea cell line bank, Seoul, Korea) were cultured in DMEM added with 10% FBS and 1% penicillin-streptomycin (Life Technologies Inc., Grand Island, NY, USA). These cells were grown at 37 °C under 5% CO_2_.

### 4.7. MTT Assay

The cells were prepared in 96-well plates. Each well contained 100,000 cells/mL in 100 μL of medium. After 24 h incubation, various doses of EEP were added in triplicate, and cells were incubated for 24 h. After 4 h of incubation with MTT solution (5 mg/mL in PBS) 20 μL, the medium was removed and purple formazan crystals was dissolved with 200 μL of DMSO. The absorbance was measured by micromultiplate reader at 540 nm.

### 4.8. Measurement of NO and Cytokines

Cells were pretreated with various dose of EEP for 1 h, and treated with/without LPS for 24 h. The nitrite accumulated in culture medium was detected as an indicator of NO production based on the Griess reaction. Culture medium (100 μL) was mixed with 100 μL of Griess reagent (equal volumes of 1% (*w*/*v*) sulfanilamide in 5% (*v*/*v*) phosphoric acid and 0.1% (*w*/*v*) naphtylethylenediamine-HCl), and incubated at room temperature for 10 min. Next, the absorbance of mixture was measured at 540 nm by a micromultiplate reader. Fresh culture medium was used as the blank in all experiments. The amount of nitrite in the samples was measured with the sodium nitrite serial dilution standard curve. Also, the levels of IL-6, IL-1β, and TNF-α in culture medium were quantified through EIA kits, following the manufacturer’s instructions.

### 4.9. Animals

The male ICR mice (5-week-old, 18–22 g) were purchased from Orient Bio (Seoul, Korea). Mice were inhabited 5/cage and were had standard laboratory chow in an animal room with 12 h dark/light cycles at a constant temperature of 20 ± 5 °C. All animal experiments were carried out under university guidelines and were approved by the ethical committee for Animal Care and Use of Kyung Hee University according as the animal protocol (KHUASP(SE)-17-148, 18/01/2018).

### 4.10. Induction of Colitis

Colitis was induced by feeding mice with drinking water containing 4% (*w*/*v*) DSS for 9 days (ad libitum). Each group was given same volume of DSS-containing water, and was monitored every day to check whether mice consumed an approximately equal volume of DSS-containing water. For each experiment, the mice were separated into 6 experimental groups, and they were treated by different material for 9 days; (1) mice drinking normal water and feeding vehicle orally (p.o.) once daily (vehicle-treated control group); (2) mice drinking DSS-containing water and feeding vehicle orally (p.o.) once daily (DSS only-treated group); (3) mice drinking DSS-containing water and receiving 75 mg/kg/day of 5-ASA orally daily (DSS plus 5-ASA group); (4–6) mice drinking DSS-containing water and receiving 50, 100 or 200 mg/kg/day of EEP orally daily (DSS plus EEP 50, 100 or 200 mg/kg group). All materials were dissolved in 0.9% saline, and the administration of each drug was started at the same time as the DSS treatment. The experiments were conducted twice with ten mice in each group.

### 4.11. Evaluation of DAI

Body weight, stool consistency, and gross bleeding were recorded daily. The DAI was determined by averaging the scores of (i) the body weight loss, (ii) stool consistency and (iii) gross bleeding. Each score was determined as follows: change in body weight loss (0: none, 1: 1–5%, 2: 5–10%, 3: 10–20%, 4: >20%), stool blood (0: negative, 1: +, 2: ++, 3: +++, 4: ++++) and stool consistency (0: normal, 1 and 2: loose stool, 3 and 4: diarrhea). The body weight loss was calculated as the percent difference between the original body weight (day 0) and the body weight on any particular day (Table 3. The mice were sacrificed at the end of the experiment, and the colons were separated from the proximal rectum, close to its passage under the pelvisternum. The colon length was measured between the ileocecal junction and the proximal rectum. The spleens were also obtained, and their weight was measured.

### 4.12. Western Blot Analysis

Colon segments or cells were resuspended in PRO-PREP^TM^ protein extraction solution (Intron Biotechnology, Seoul, Korea) and incubated for 30 min at 4 °C. Cell debris was removed by microcentrifuge (4 °C, 15,000 rpm, 30 min) and then the protein concentration of supernatants was ascertained by Bio-rad protein assay reagent, following the manufacturer’s instruction. The 20–30 µg of protein was fractionated by 8~15% SDS PAGE, and transferred to a PVDF. After incubating for 1 h with 5% skim milk in Tween 20/Tris-buffered saline (T/TBS) at 20 °C, the membranes were incubated with primary antibody at 4 °C for overnight. Membranes were rinsed thrice with T/TBS and incubated with secondary antibody for 2 h at 25 °C, rewashed thrice with T/TBS. Blots were developed using enhanced chemiluminescence detection agents (Amersham, Buckinghamshire, UK).

### 4.13. Histopathology

The resected large intestine was grossly examined for mucosal defects, hemorrhage, or ulcerative lesions, and it was immediately fixed in 4% paraformaldehyde overnight and embedded in paraffin. Histopathology and immunohistochemical staining were performed by Korea Experimental Pathology Inc. (Gyeonggi-do, Korea). For the histopathological analysis, tissue sections were made from the representative region of the large intestine using conventional tissue preparation methods, and the samples were viewed under a light microscope (400×) after H&E staining. The immunohistochemical detection of p-p65 and p-STAT3 was carried out using the avidin-biotin-DAB complex method on paraffin sections. Briefly, the samples were incubated overnight at 4 °C with primary monoclonal antibodies against p-p65 and p-STAT3 (Santa Cruz, diluted 1:50), and then a biotin conjugated goat anti-rabbit secondary antibody (Vector Laboratories, Burlingame, CA, USA, diluted 1:250) and, subsequently, streptavidin conjugated with horseradish peroxidase (Vector Laboratories, diluted 1:250) were applied. A DAB peroxidase substrate (Vector Laboratories) was utilized for visualization, and the specimens were counterstained with hematoxylin (Sigma Chemical). For determination of MPO accumulation, colonic tissues were thawed and homogenized in lysis buffer (50 mM potassium phosphate (pH 6.0, 0.5% hexadecyltrimethylammonium bromide)), centrifuged at 15,000× *g* for 10 min, and then supernatant was assayed by the *o*-dianisidine method.

### 4.14. Quantitative Real-Time Reverse-Transcriptase Polymerase Chain Reaction (qRT-PCR)

The total cellular RNA was extracted by Easy Blue^®^ kits (Intron Biotechnology, Seoul, Korea). From each sample, 500 ng of RNA was reverse-transcribed by TOPscript™ RT Dry MIX, and quantitative real-time PCR amplification was carried out by the incorporation of TB Green^TM^ Premix Ex Taq (TaKaRa) to detect IL-6, IL-1β, TNF-α, iNOS, COX-2, F4/80, and β-actin mRNA expression. The oligonucleotide primers are listed in Table 4. A dissociation curve analysis of IL-6, IL-1β, TNF-α, iNOS, COX-2, F4/80, and β-actin showed a single peak for each. The mean Ct of the gene of interest was calculated from triplicate measurements and was normalized with the mean Ct of a control gene, β-actin.

### 4.15. Statistical Analysis

Data are represented as the mean ± SDs (*n* = 10) in vivo and mean ± SD of triplicate experiments in vitro. Statistical significances were identified using ANOVA and Dunnett’s post *hoc* test, and *p*-values of less than 0.05 were reputed statistically significant.

## 5. Conclusions

EEP exhibits protective and anti-colitis effects in DSS-induced mice model and anti-inflammatory properties in LPS-induced RAW 264.7 macrophage cells by suppressing phosphorylation of NF-κB and STAT3. Therefore, EEP might be useful as a complementary and alternative supplement to prevent IBD.

## Figures and Tables

**Figure 1 ijms-20-00177-f001:**
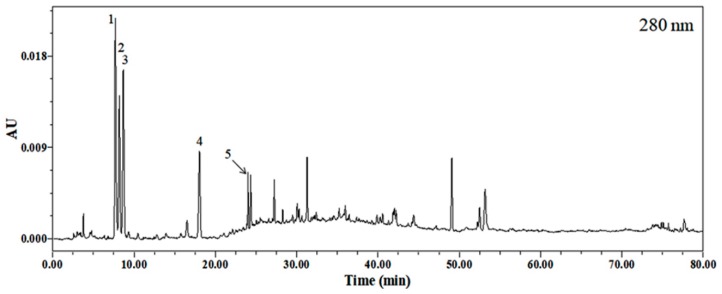
HPLC chromatogram of *Persea americana* Mill. (avocado) extract detected at 280 nm.

**Figure 2 ijms-20-00177-f002:**
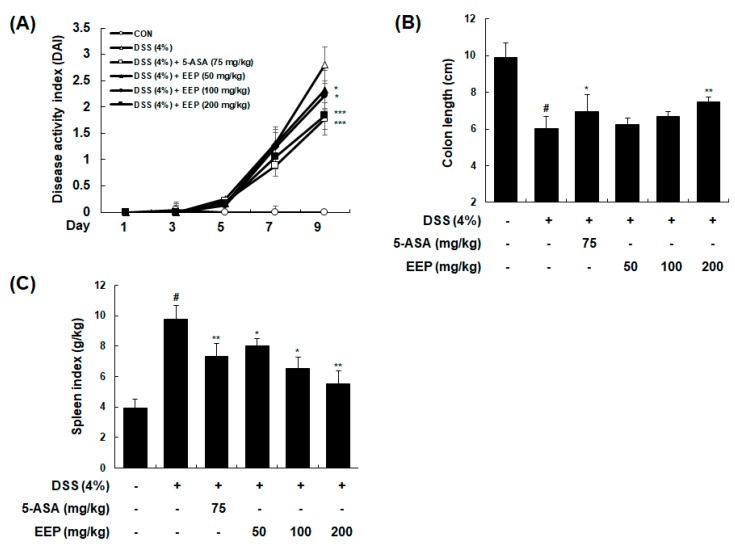
Effects of EEP on clinical signs of DSS-induced colitis mice. Mice were administered 4% DSS in drinking water (ad libitum) for 9 days with or without EEP (50, 100, or 200 mg/kg/day p.o.). 5-ASA (75 mg/kg/day p.o.) was used as a positive control. On day 9, mice were sacrificed and their colons and spleens were collected. (**A**) Changes in DAI levels were evaluated daily throughout the 9 days of administration period. (**B**) Length of colons and (**C**) weight of spleens were measured. Values are presented as means ± SDs (*n* = 10). # *p* < 0.05 compared with vehicle-treated control group, and * *p* < 0.05, ** *p* < 0.01, *** *p* < 0.001 compared with DSS-induced colitis group.

**Figure 3 ijms-20-00177-f003:**
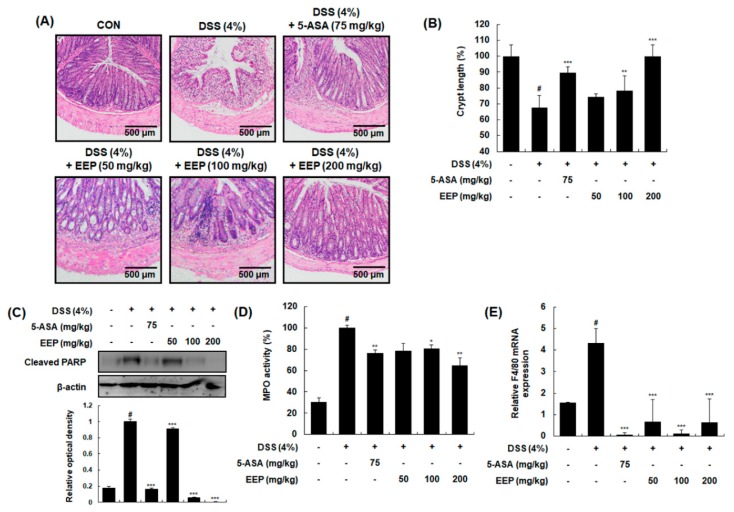
Effects of EEP on histological changes in DSS-induced colitis mice. (**A**) Representative sections of colonic tissues from mice administrated 4% DSS in drinking water (ad libitum) for 9 days with or without EEP. 5-ASA (75 mg/kg/day p.o.) was used as a positive control. Histological changes were determined by H&E staining. (**B**) Changes in crypt length were measured. (**C**) Whole proteins were prepared from DSS-exposed colonic tissues and subjected to Western blotting to determine expression level of cleaved PARP. β-Actin was used as an internal control. (**D**) Colon segments from mice were used to determine myeloperoxidase (MPO) levels. (**E**) Total RNAs were prepared from DSS-exposed colonic tissues and used for qRT-PCR. Using specific primers, mRNA expression levels of F4/80 were determined and normalized against β-actin. Values are presented as means ± SDs (*n* = 10). # *p* < 0.05 compared with the vehicle-treated control group, and * *p* < 0.05, ** *p* < 0.01, *** *p* < 0.001 compared with the DSS-induced colitis group.

**Figure 4 ijms-20-00177-f004:**
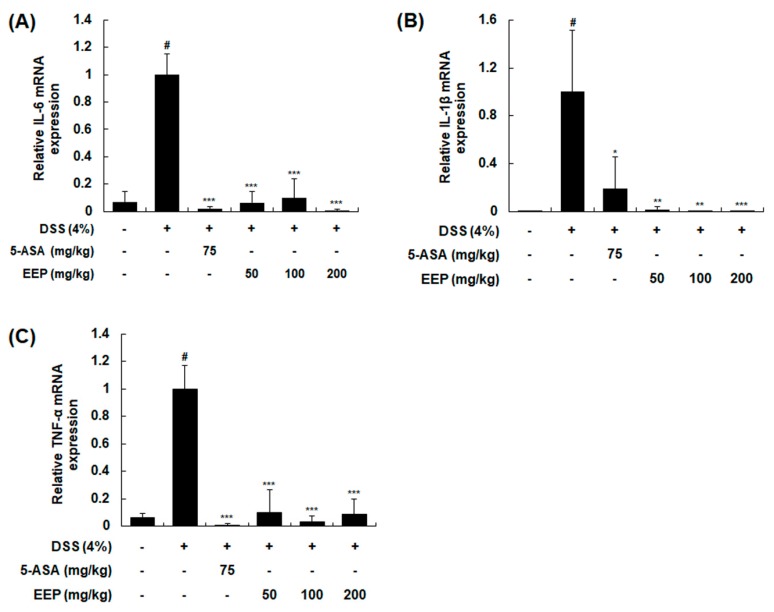
Effects of EEP on mRNA expression levels of pro-inflammatory cytokines in DSS-induced colitis mice. Colonic tissues were obtained after 9 days of DSS treatment and total RNAs were prepared for qRT-PCR. mRNA expression levels of IL-6 (**A**), IL-1β (**B**), and TNF-α (**C**) were determined using specific primers and normalized against β-actin. Values are presented as means ± SDs (*n* = 10). # *p* < 0.05 compared with the vehicle-treated control group, and * *p* < 0.05, ** *p* < 0.01, *** *p* < 0.001 compared with the DSS-induced colitis group.

**Figure 5 ijms-20-00177-f005:**
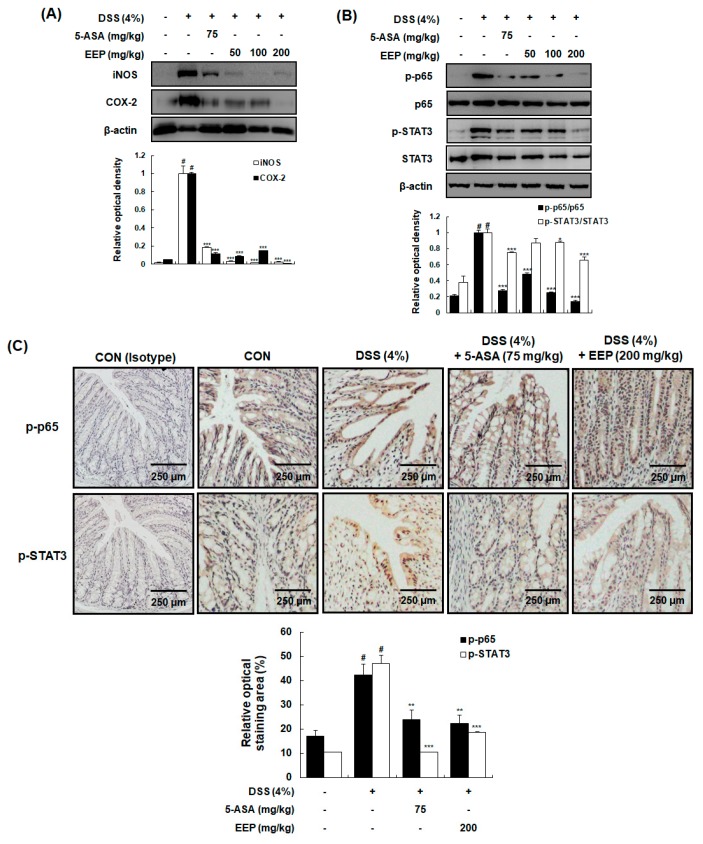
Effects of EEP on expression of iNOS and COX-2 and activation of NF-κB and STAT3 in DSS-induced colitis mice. (**A**,**B**) Colonic tissues were obtained after 9 days of DSS treatment and whole proteins were prepared for Western blotting to detect protein expression levels of iNOS, COX-2, p-p65, p65, p-STAT3, and STAT3. β-Actin was used as an internal control. (**C**) Representative colonic sections from mice were immunostained for p-p65 and p-STAT3. The normal rabbit IgG antibody and the normal mouse IgG_2b_ antibody were used as isotype control for p-p65 and p-STAT3, respectively. Values are presented as means ± SDs (*n* = 10). # *p* < 0.05 compared with vehicle-treated control group, and * *p* < 0.05, ** *p* < 0.01, *** *p* < 0.001 compared with DSS only-treated group.

**Figure 6 ijms-20-00177-f006:**
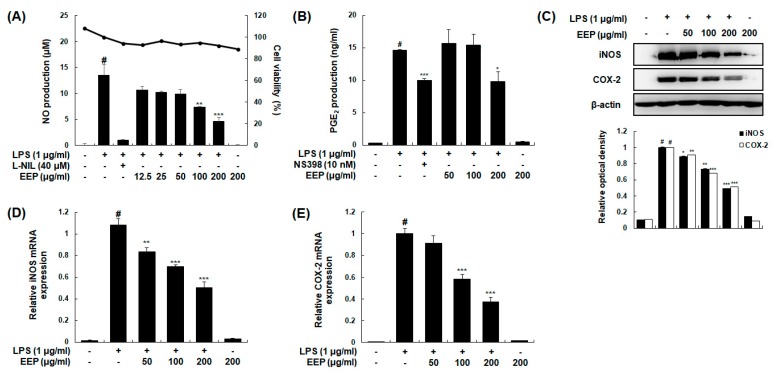
Effects of EEP on LPS-induced production of NO and PGE_2_ and expression of iNOS and COX-2 in RAW 264.7 macrophages. Following treatment with or without indicated dose of EEP for 1 h, cells were treated with or without LPS (1 μg/mL) for 24 h. Control cells were not treated with EEP or LPS. (**A**,**B**) Production of NO and PGE_2_ was determined by Griess reaction and EIA kits, respectively. l-NIL (40 µM) and NS-398 (10 nM) were used as positive control for NO and PGE_2_ production, respectively. (**C**) Total cellular proteins were obtained from cells stimulated with LPS (1 μg/mL) for 24 h with or without EEP (50, 100, or 200 μg/mL). Protein expression levels of iNOS and COX-2 were detected by Western blotting. β-Actin was used as an internal control. (**D**,**E**) Total RNAs were prepared from cells stimulated with LPS for 6 h in presence or absence of EEP (50, 100, or 200 μg/mL). mRNA levels of iNOS and COX-2 were determined by qRT-PCR using specific primers. Results were normalized against β-actin. Experiments were performed independently three times and similar results were obtained. Values shown are the means ± SDs of three independent experiments. # *p* < 0.05 vs. control group; * *p* < 0.05, ** *p* < 0.01, *** *p* < 0.001 vs. LPS-stimulated group.

**Figure 7 ijms-20-00177-f007:**
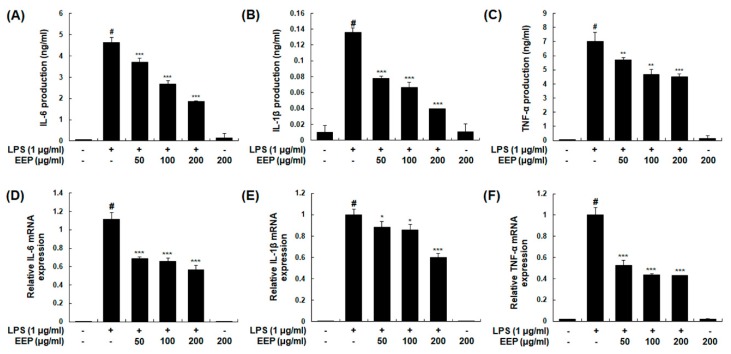
Effects of EEP on production and mRNA expression of pro-inflammatory cytokines in RAW 264.7 macrophages. (**A**–**C**) Following treatment with EEP (50, 100, or 200 μg/mL) for 1 h, cells were stimulated with LPS (1 μg/mL) for 24 h. Production of IL-6, IL-1β, and TNF-α in the culture media was determined using EIAs kits. Control cells were not treated with LPS or EEP. (**D**–**F**) Total RNAs were prepared for qRT-PCR analysis of IL-6, IL-1β, and TNF-α in cells stimulated with LPS (1 μg/mL) for 2 h (TNF-α) or 6 h (IL-6 and IL-1β) with or without treatment with EEP. mRNA levels of IL-6, IL-1β, and TNF-α were determined using gene specific primers. Results were normalized against β-actin. Experiments were repeated three times and similar results were obtained. Values shown are means ± SDs of three independent experiments. # *p* < 0.05 vs. control group; * *p* < 0.05, ** *p* < 0.01, *** *p* < 0.001 vs. LPS-stimulated group.

**Figure 8 ijms-20-00177-f008:**
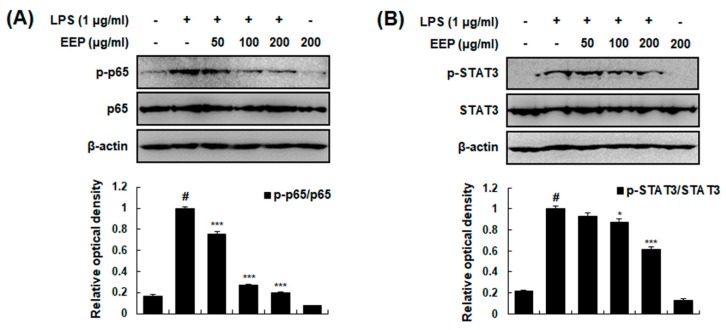
Effects of EEP on activation of NF-κB and STAT3 in RAW 264.7 macrophages. (**A**,**B**) Total cellular proteins were obtained from cells stimulated with LPS (1 μg/mL) for 15 min (p-p65) or 2 h (p-STAT3) with or without indicated dose of EEP and analyzed by Western blotting. Levels of p-p65, p65, p-STAT3, and STAT3 were determined using specific antibodies. β-Actin was used as an internal control. Experiments were repeated three times and similar results were obtained. Values shown are means ± SDs of three independent experiments. # *p* < 0.05 vs. control group; * *p* < 0.05, *** *p* < 0.001 vs. LPS-stimulated group.

**Table 1 ijms-20-00177-t001:** Retention time (Rt), precursor ion, molecule formula, and ultraviolet maxima (λmax) of peaks identified in ethanol extract of *P. americana* Mill. (EEP).

Compound	Rt (min)	Precursor Ion (*m*/*z*)	Molecular Formula	λmax (nm)
1. Homocysteine	7.70	136.04344 [M + H]^+^	C_4_H_9_NO_2_S	264
2. Guanine	8.20	152.04841 [M + H]^+^	C_5_H_5_N_5_O	253
3. Cytidine	8.68	244.09982 [M + H]^+^ 112.02048 [M-glu + H]^+^	C_9_H_13_N_3_O_5_	215, 279
4. Uridine	18.03	245.08380 [M + H]^+^ 267.06869 [M + Na]^+^	C_9_H_12_N_2_O_6_	261
5. Guanosine	24.02	284.10544 [M + H]^+^ 152.04486 [M-glu + H]^+^	C_10_H_13_N_5_O_5_	253

**Table 2 ijms-20-00177-t002:** Total polyphenol and flavonoid contents of EEP.

	Total Polyphenol	Total Flavonoid
Contents (mg/g)	4.52 ± 0.01	1.56 ± 0.05

**Table 3 ijms-20-00177-t003:** Evaluation of disease activity index (DAI).

DAI Score	Weight Loss (%)	Stool Consistency	Occult/Gross Bleeding
0	None	Normal	Normal
1	1–5	Loose stools	Hemoccult positive
2	5–10		
3	10–20		
4	>20	Diarrhea	Gross bleeding

**Table 4 ijms-20-00177-t004:** List of primers.

Gene	Sequence	
*IL-6*	Forward	GAGGATACCACTCCCAACAGACC
	Reverse	AAGTGCATCATCGTTGTTCATACA
*IL-1β*	Forward	ACCTGCTGGTGTGTGACGTT
	Reverse	TCGTTGCTTGGTTCTCCTTG
*TNF-α*	Forward	AGCACAGAAAGCATGATCCG
	Reverse	CTGATGAGAGGGAGGCCATT
*iNOS*	Forward	AATGGCAACATCAGGTCGGCCATCACT
	Reverse	GCTGTGTGTCACAGAAGTCTCGAACTC
*COX-2*	Forward	GGAGAGACTATCAAGA TAGT
	Reverse	ATGGTCAGTAGACTTTTACA
*F4/80*	Forward	AGGACTGGAAGCCCATAGCCAA
	Reverse	GCATCTAGCAATGGACAGCTG
*β-actin*	Forward	ATCACTATTGGCAACGAGCG
	Reverse	ATCACTATTGGCAACGAGCG

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
