# Peer review of "Anti-Colitic Effects of Ethanol Extract of *Persea americana* Mill. through Suppression of Pro-Inflammatory Mediators via NF-κB/STAT3 Inactivation in Dextran Sulfate Sodium-Induced Colitis Mice"

_ijms, 2019, doi:10.3390/ijms20010177_

Reviewer 1 Report

This article is very interesting and innovative. However, to demonstrate the growing scientific interest for the plant, I recommend to include in the paper a graphic of number publications/years. Moreover, it would be useful to make a Folin assay of the extract, to demonstrate that the biological effects of the EEP are also dependent of its polyphenol content. Please, motivate the choice to use the ethanol/water as solvent to obtain EEP. What is the reason for using RAW 264.7 instead of similar but differentiated cells, which are closer to the in vivo model? In future, do you plan to use in animal models a mitofagic inhibitor, such as cyclosporine and to evaluate the expression and mitochondrial recruitment of PINK-1 and Parkin in the cells?  Are there correlations between the doses in vivo and in vitro in your experiments? As minor point, please correct mg/Kg in mg/ml in lanes 221 and 222.

Author Response

Thanks for your comments. We uploaded our response to your comments.

Reviewer 2 Report

This manuscript demonstrates that ethanol extract of P.americana (EEP) could improve clinical signs and histological characteristics of DSS-induced colitis mice. Moreover, EEP decreased the expression levels of iNOS, COX-2, pro-inflammatory cytokines, and phosphorylation of NF-κB and STAT3 in DSS-exposed colitis tissues. Furthermore, authors found EEP could reduce iNOS, COX-2, pro-inflammatory cytokine expression and the level of NF-κB and STAT3 phosphorylation in LPS-stimulated RAW 264.7 macrophages.  The results are very interesting. The paper could be greatly improved if note is taken of the following points:

1. In Fig.5, epithelial cells are stained by p-p65 antibody and p-STAT3 antibody. Therefore, authors should examine the effect of EEP on NF-κB and STAT3 phosphorylation in epithelial cells.

 2. Did authors use isotype control antibody in this immunohistochemical study? This reviewer thinks authors should show the picture using isotype control antibody in this immunohistochemical analysis. Moreover, authors should mention the information about isotype control antibody in Materials and Methods.

 3. Authors used 200μg/ml EEP in vitro study. Did authors examine the concentration of EEP in DSS and EEP-exposed colitis tissues? Moreover, authors should show where macrophages exist in DSS-exposed colitis tissues by immunohistochemical study because authors used macrophages in vitro study.

 4. Authors examine the effect of EEP on iNOS, COX-2, and pro-inflammtory cytokine expression in LPS-stimulated macrophages. Did macrophage express iNOS, COX-2, pro-inflammtory cytokine in DSS-exposed colitis tissues? Please explain in Discussion.

Author Response

(The authors gave the same response as above.)

Reviewer 3 Report

This is an interesting data-rich manuscript. The experiments are generally well designed and the manuscript are well written.  Only one comment: Figure 1 is not meaningful. The chemicals listed do not have much anti-inflammatory effect. 

Author Response

Thanks for your comments. We uploaded our response to your comments.

Response to Reviewer 3 Comments Reviewer 3 This is an interesting data-rich manuscript. The experiments are generally well designed and the manuscript are well written. Point 1. Only one comment: Figure 1 is not meaningful. The chemicals listed do not have much anti-inflammatory effect. Response 1: Thank you for your critical comment. We added below paragraph in results and discussion parts of revised manuscript. ‘Nucleosides and nucleotides play important roles in many cellular functions including inflammation regulation by acting as signaling molecules [1]. Homocysteine has been reported to show anti-inflammatory properties in a hypercholesterolemic rat model in vivo [2]. Uridine also has been demonstrated to reduce inflammatory cell infiltration and cytokine production in sephadex-induced lung inflammation [1]. In addition, guanosine exhibits anti-inflammatory activity by associating the heme oxygenase-1 (HO-1) signaling in hippocampal astrocytes [3]. These reports suggest that nucleosides from EEP are active components for anti-colitic and anti-inflammatory properties. Moreover, we determined the total polyphenol and total flavonoid contents of EEP to support the anti-inflammatory activity of EEP. EEP has total phenol and total flavonoid contents of 4.52 ± 0.01 mg/g and 1.56 ± 0.05 mg/g, respectively.’ References 1. Evaldsson, C.; Ryden, I.; Uppugunduri, S. Anti-inflammatory effects of exogenous uridine in an animal model of lung inflammation. Int Immunopharmacol 2007, 7, 1025-1032, http://10.1016/j.intimp.2007.03.008 2. Pirchl, M.; Ullrich, C.; Sperner-Unterweger, B.; Humpel, C. Homocysteine has anti-inflammatory properties in a hypercholesterolemic rat model in vivo. Mol Cell Neurosci 2012, 49, 456-463, http://10.1016/j.mcn.2012.03.001 3. Bellaver, B.; Souza, D.G.; Bobermin, L.D.; Goncalves, C.A.; Souza, D.O.; Quincozes-Santos, A. Guanosine inhibits LPS-induced pro-inflammatory response and oxidative stress in hippocampal astrocytes through the heme oxygenase-1 pathway. Purinergic Signal 2015, 11, 571-580, http://10.1007/s11302-015-9475-2

Reviewer 4 Report

The manuscript is very interesting and is well-structured in general. The Authors performed both in vivo and in vitro studies to reveal the anti-colitic effects of ethanol extract of Persea americana Mill. Moreover, the potential signalling pathway underlying these effects was demontrated. In my opinion, this manuscript could be interesting for a reasonable number of scientists. I suggest acceptance of this research article in its present form.

Author Response

Thanks for your comment.

Round  2

Reviewer 1 Report

Thank you for answering my questions.

Reviewer 2 Report

This reviewer thinks authors corrected the manuscript as this reviewer suggested. So, this reviewer thinks the manuscript is ready for publishing in IJMS.